# Emotion Recognizing by a Robotic Solution Initiative (EMOTIVE Project)

**DOI:** 10.3390/s22082861

**Published:** 2022-04-08

**Authors:** Grazia D’Onofrio, Laura Fiorini, Alessandra Sorrentino, Sergio Russo, Filomena Ciccone, Francesco Giuliani, Daniele Sancarlo, Filippo Cavallo

**Affiliations:** 1Clinical Psychology Service, Health Department, Fondazione IRCCS Casa Sollievo della Sofferenza, San Giovanni Rotondo, 71013 Foggia, Italy; f.ciccone@operapadrepio.it; 2Department of Industrial Engineering, University of Florence, 50121 Florence, Italy; laura.fiorini@unifi.it (L.F.); alessandra.sorrentino@unifi.it (A.S.); filippo.cavallo@unifi.it (F.C.); 3Information and Communication Technology, Innovation & Research Unit, Fondazione IRCCS Casa Sollievo della Sofferenza, San Giovanni Rotondo, 71013 Foggia, Italy; s.russo@operapadrepio.it (S.R.); f.giuliani@operapadrepio.it (F.G.); 4Complex Unit of Geriatrics, Department of Medical Sciences, Fondazione IRCCS Casa Sollievo della Sofferenza, San Giovanni Rotondo, 71013 Foggia, Italy; d.sancarlo@operapadrepio.it

**Keywords:** human-robot interaction, acceptability, non-verbal cues and expressiveness, monitoring of behaviorand internal states of humans

## Abstract

Background: Emotion recognition skills are predicted to be fundamental features in social robots. Since facial detection and recognition algorithms are compute-intensive operations, it needs to identify methods that can parallelize the algorithmic operations for large-scale information exchange in real time. The study aims were to identify if traditional machine learning algorithms could be used to assess every user emotions separately, to relate emotion recognizing in two robotic modalities: static or motion robot, and to evaluate the acceptability and usability of assistive robot from an end-user point of view. Methods: Twenty-seven hospital employees (M = 12; F = 15) were recruited to perform the experiment showing 60 positive, negative, or neutral images selected in the International Affective Picture System (IAPS) database. The experiment was performed with the Pepper robot. Concerning experimental phase with Pepper in active mode, a concordant mimicry was programmed based on types of images (positive, negative, and neutral). During the experimentation, the images were shown by a tablet on robot chest and a web interface lasting 7 s for each slide. For each image, the participants were asked to perform a subjective assessment of the perceived emotional experience using the Self-Assessment Manikin (SAM). After participants used robotic solution, Almere model questionnaire (AMQ) and system usability scale (SUS) were administered to assess acceptability, usability, and functionality of robotic solution. Analysis wasperformed on video recordings. The evaluation of three types of attitude (positive, negative, andneutral) wasperformed through two classification algorithms of machine learning: k-nearest neighbors (KNN) and random forest (RF). Results: According to the analysis of emotions performed on the recorded videos, RF algorithm performance wasbetter in terms of accuracy (mean ± sd = 0.98 ± 0.01) and execution time (mean ± sd = 5.73 ± 0.86 s) than KNN algorithm. By RF algorithm, all neutral, positive and negative attitudes had an equal and high precision (mean = 0.98) and F-measure (mean = 0.98). Most of the participants confirmed a high level of usability and acceptability of the robotic solution. Conclusions: RF algorithm performance was better in terms of accuracy and execution time than KNN algorithm. The robot was not a disturbing factor in the arousal of emotions.

## 1. Introduction

The concept of emotion has a place in evolutionary history and it even predates the rise of human beings [1]. This piece of knowledge corroborates the idea that some emotions are innate in humans rather than socially constructed [2]. While processing emotions may come naturally to most people, computers, on the other hand, have struggled with the execution of this concept for decades. Much research has been gathered in fields like computer vision and machine learning in the effort to use computers to accurately classify emotions portrayed by humans. This effort includes analyzing human speech, gestures, and facial expressions.

Many areas offer potential for the application of automated human facial expression recognition. Such fields include education, marketing, security, and medicine. Essentially, wherever a human is present to evaluate emotion, a computer can also aid in the analysis.

In the future, assistive robots maybe components of our daily lives exhibiting a high level of social interaction [3]. In the human–robot interaction (HRI), assistive robots are perceived associal actors evoking mental patterns characteristic of the human–human interaction [4].

A person’s attitude towards social interaction is expressed by a series of social signals that convey information concerning feelings, mental state, and other individual traits (e.g., voice quality, posture, gaze) [5]. Furthermore, social relationships are complex and take in several indicators (e.g., emotions, language, tone, facial expression, and body posture).

In order to rise above HRI gaps, new robots need to incorporate cognitive models and create a real vision of how human interaction models employ [6], identifying the user behavior and its changing modality during the interaction and modelling the robot behavior consequently. De Carolis et al. [7] described NAO robot capable to simulate empathic behavior based on the user’s recognized emotion through speech prosody and facial expression with a higher capability to understand emotions (4.11 vs. 3.35) than ability to feel emotions.

TheiCub robot wasalso provided with affective and interactive prompts based on facial expressions [8]: the authors reported that iCub’s internal dynamics was perceived by users more positively with adaptive vs. non-adaptive robot behavior with significant differences found both over the sessions [F(1, 22) = 7.87, *p* = 0.01] and for interaction [F(1, 22) = 5.27, *p* = 0.03].

Of the key factors of robots and human interaction is the robot human-like head. As a result, a person could more naturally convey theirthoughts to the robot, which consequently could helpthem to express their internal state [9,10]. However, several challenges persist in order to achieve research outcomes.

The first important question concerns the modalities needed to sense the emotional state of people by the robot. Secondly, there is the problem of modelling the interaction between human and robot on the emotional level.

Emotion recognition skills should be present in real social service robots moving in an open world, closelyrelated to a group of people to perform specific tasks.

Some experimenters believe that emotion recognition in robots should be used to collect feedback on user comportment of robots and also adapt it consequently. Mei and Liu [11] assembled a robotic grabber with two cameras, one to detect objects and the other to observe the user’s face. Based on the emotion expressed by the user during the process, the robot decided to capture or avoid the displayed object. Chumkamon et al. [12] have developed a robot capableof sympathizing with the emotion expressed by the user. Interestingly, unsupervised learning was used in both of these cases. Nho et al. [13] estimated a cleaning robot’s actions by using hand gestures to fine-tune its internal emotional model. In these cases, the user showed an emotion caused directly by the robot’s actions, so the emotion was related to a specific action and not to the general environment. Hence, a robot could also be tutored to act directly based on the user’s emotional feeling, therefore unconnected to the robot’s actions. The robot could use it to fine- tune internal parameters, for example not to act too invasively in the task performed. Roning et al. [14] developed a remarkable illustration of an independent agent carrying in the real world; their Minotaurus robot could recognize people through sight, discover their mood, and move around intelligent ambient and take information from it.

Another group of experimenters supposed that robots should have an internal model for feelings to regulate their actions. Jitviriya et al. [15] used a self-organizing map (SOM) to classify eight inner feelings of a ConBe robot. Based on the difficulty of catching the ball, the robot tuned its actions and emotional expression when it caughtthe ball held by a person. Van Chien et al. [16] have developed a humanoid robotic platform that expresses its internal feelings through changes in the way it walks.

The design of a social robot depends on the operation and the set of contexts it manages. The presence of a head in the robot’s body can clearly be an effective way for robots to express feelings, although its design can be relatively grueling and constrained.

Emotion recognition in robotics is important for several reasons, not only for the pure interaction between human and robot but also for bioengineering operations. A substantial part of the investigativework in this area concerns the treatment of children on theautism spectrum or seniors with mild cognitive impairments [16]. In this case, robots are used as tools to show the feelings that children need to showduring the therapy session, as they can communicate with robots more fluently than with humans. Additionally, they discover the emotion expressed by the patients and try to classify it according to the correct degree of performance [17,18,19,20,21]. A physiotherapist is always present to cover the work and judge the categories made by the robots, as these results are notintended to replace human work [16]. In fact, in these operations, robots act as simple tools to help workers and are more known as support robots. In similar operations, a robot is awaited to give only an objective evaluation that complements the individual evaluation of the physiotherapists. It does notrequire emotion recognition or environment mindfulness to shape one’s actions as it has no real emotion. Otherwise, the robot could be used to identifya person’s mood inside their home and try to relieve them of a stressful condition. An interesting sample comes from Han et al. [22]. Their robot recognized emotions in elderly people and changed the parameters of an intelligent environmental context to try to induce happiness. In an alternate case, it could be used to discover the onset of dementia or mild cognitive impairment.

Since facial detection and recognition algorithms are compute-intensive operations, it needs to identify methods that can parallelize the algorithmic operations for large-scale information exchange in real time. Emergence of low-cost graphic processing units (GPU), many-core architectures, and high-speed Internet connectivity has provided sufficient communication resources to accelerate facial detection and recognition [23].

In a recent study, a conditional-generative-adversarial-network-based (CGAN- based) framework was used to drop intra-class variances through management of facial expressions and to learn generative and discriminative representations [24]. The face image wasconverted into a prototypic shape of facial expression by the generator G with an accuracy of 81.83%. In other studies, a convolutional neural network (CNN)-based multi-task model was designed for gender classification, smile discovery, and emotion recognition with an accuracy of 71.03% [25] and performance accuracies of 76.74% and 80.30% with the integration of k-nearest neighbor (KNN) technique [26].

Additionally, CNN can be trained for the face identification task. Thus, random forest (RF) classifiers were learned to predict an emotion score using an available training set with an accuracy of 75.4% [27].

The present study addressed a main challenge: how can the principles of engineering and mathematics be applied to create a system that can recognize emotions in given facial images?

The specific aims of the present study were:to identify if traditional machine learning algorithms could be used to evaluate each user’s emotions independently (Intra-classification task);to compare emotion recognitionin two types of robotic modalities: static robot (which does not perform any movement) and motion robot (whichperforms movements in concordant with the emotions elicited); andto assess the acceptability and usability of assistive robot from the end-user point of view.

## 2. Materials and Methods

### 2.1. Robot Used for Experimentation

For this studyphase, Pepper robot [28] was used (Figure 1).

Pepper is the world’s first social humanoid robot able to recognize faces and basic human emotions. It was optimized for human interaction and is able to engage with people through conversation and its touch screen.

It has six main characteristics, as shown below:Twenty freedom degrees for natural and expressive movements.Perception components to identify and interact with the person talking to it.Speech recognition and dialogue available in 15 languages.Bumpers, infrared sensors, 2D and 3D cameras, sonars for omnidirectional and autonomous navigation, and an inertial measurement unit (IMU).Touch sensors, LEDs, and microphones for multimodal interactions.Open and fully programmable platform.

Standing 120 cm tall, Pepper has no trouble perceiving its environment and entering into a conversation when it sees a person. The touch screen on its chest displays content to highlight messages and support speech. Its curvy design ensures danger-free use and a high level of acceptance by users.

### 2.2. Recruitment

This study fulfilled the Declaration of Helsinki [29], guidelines for Good Clinical Practice, and the Strengthening the Reporting of Observational Studies in Epidemiology guidelines [30]. The approval of the study for experiments using human subjects was obtained from the local Ethics Committee on human experimentation (Prot. N. 3038/01DG). Written informed consent for research was obtained from each participant.

Participants were recruited in July 2020 in Casa Sollievo della Sofferenza Hospital, San Giovanni Rotondo, Italy.

In total, 30 hospital employees were screened for eligibility.

Inclusion criteria were:No significant neuropsychiatric symptoms evaluated by neuropsychiatric inventory (NPI) [31]: through use of LabView, an interface to upload data relating to the NPI has been created in order to run the calculation of the uploaded data;No significant visual or hearing loss; andNo cognitive impairment evaluated by mini mental state examination (MMSE) [32]: MMSE score ≥ 27.

Exclusion criteria were:No completed and signed informed consent;Incomplete acceptability and usability assessment; andRecorded video that is not properly visible.

### 2.3. Experimentation Protocol

The experiment was run by a showing of International Affective Picture System (IAPS) [33] capable ofeliciting emotions. IAPS is a database of pictures designed to provide a standardized set of pictures for studying emotion and attention that has been widely used in psychological research. The IAPS was developed by the National Institute of Mental Health Centre for Emotion and Attention at the University of Florida. It is the essential property of the IAPS that the stimulus set is accompanied by a detailed list of average ratings of the emotions elicited by each picture. This enables other researcher to select stimuli eliciting a specific range of emotions for their experiments when using the IAPS. The process of establishing such average ratings for a stimulus set is also referred to as standardization by psychologists.

The normative rating procedure for the IAPS is based on the assumption that emotional assessments can be accounted for by the three dimensions’ valence, arousal, and dominance. Thus, participants taking part in the studies that are conducted to standardize the IAPS are asked to rate how pleasant or unpleasant, calm or excited, and controlled or in-control they felt when looking at each picture. A graphic rating scale, the Self-Assessment Manikin (SAM) [34], is used for this rating procedure.

Of this database, 60 pictures with 20 positive, negative, or neutral valence, respectively, were selected.

The positive valence is characterized by emotional states such as joy, happiness, affection, and amusement. The negative valence is characterized by behaviors such as irritation, disappointment, impatience, and annoyance.

The experiment was performed with Pepper robot in two modalities: static and active.

About the experimental phase with Pepper in the active modality, a concordant mimic was programmed according to the image types (positive, negative, and neutral): the robot performed an action according to the valence of each image shown (i.e., it smiled if a picture of a smiling child was shown). In Table 1, the actions performed by the robot according to positive or negative images is shown. The actions of the robot varied in order to avoid the repetition of the same movement in close proximity.

The participants were randomly assigned to one of two groups according to static (static robot group) and active (accordant motion group) modalities of the robot.

During the experiment, the pictures were shown over the tablet mounted on the robot chest. During the session, the user was sitting in front of the robot (Figure 2). It was achieved by developing an ad-hoc web interface in which each picture was shown for 7 s.

The experiment lasted about 20–25 min for participant. All 60 pictures did not follow a specific layout, but they were shown in the same order to all participants.

For each picture, the participants were called to perform a subjective evaluation about their perceived emotional experience (emotive feedback) by SAM directly in the tablet.

The SAM quantifies subjective emotional states using sequences of humanoid figures that reproduce the different gradations of the three fundamental dimensions of evaluation (valence, activation, and control). For the hedonic valence, the figures vary from being happy and smiling to being unhappy and frowning. For arousal, the figures vary from being relaxed and asleep to being aroused with open eyes. For control, the figures vary from being wide and in control to being small and out of control. The experimenter, with each image presented, had the task to compile a grid of coding composed of six macro categories regarding the movements of the face (facial expressions such as movements of the head and the look) and the movements of the body (posture, hand gestures, gestures, of self-contact). The coding grid was created thanks to the contributions by coding systems specialized in detecting non-verbal behavior: Facial Action Coding System (FACS) [35] and Specific Affective Coding System (SPAFF) [36]. After the participants used the robotic solution, the Almere model questionnaire (AMQ) [37] and system usability scale (SUS) [38] were administered in order to assess the acceptability, usability, and functionality of the robotic solution by participants.

AMQ is a questionnaire using a 5-point Likert scale ranging from 1 to 5 (totally disagree—disagree—don’t know—agree—totally agree, respectively) designed primarily to measure users’ acceptance toward socially assistive robots. The questionnaire focuses on the following 12 constructs: (1) anxiety; (2) attitude toward technology; (3) facilitating conditions; (4) intention to use; (5) perceived adaptiveness; (6) perceived enjoyment; (7) perceived ease of use; (8) perceived sociability; (9) perceived usefulness; (10) social influence; (11) social presence; and (12) trust. The results are expressed as the average of each construct.

SUS is a tool for measuring usability. It consists of a 10-item questionnaire with five response options for respondents: strongly agree to strongly disagree. The participant’s scores for each question are converted to a new number, added together, then multiplied by 2.5 to convert the original scores of 0–40 to 0–100. A SUS score above 68 would be considered above average confirming that the system is useful and easy to use.

### 2.4. Data Analysis

The interaction with the Pepper robot wasassessed by automatically analyzing the recorded videos achieved by an external camera.

Videorecordings (front video shooting) of the user’s interaction with the robotic solution were evaluated and coded using the behavioral parameters in Table 2.

Specifically, the features wereextracted from each frame for all videos. The entiredata analysis process is shown in Figure 3. The extracted features werecollected in a unimodal dataset. Moreover, the attitude evaluation wasperformed on the dataset, using the raw representation of each instance.

#### 2.4.1. Feature Extraction

The recorded video wasthen analyzed to extract the visual features. At the end of this process, a dataset wasobtained containing the data extracted from the video (unimodal).

As described in [3], the visual features of interest wereextracted from each image frame (sampled at 30 Hz) using an open source data processing tool: OpenFace [39].

The OpenFace toolkit wasused to analyze the user’s facial behavior. In detail, we used the OpenFace toolkit [40] to estimate gaze, eye reference points, head pose, face reference point, and facial action units (AU)—total characteristics = 465, which are commonly used to evaluate emotions in affective processing. As described in [30], the OpenFace model uses a convolutional experts constrained local model (CE-CLM) which is composed of point distribution model (PDM), which detects changes in the shape of landmarks and patch experts thatlocally model variations in the appearance of each landmark. CE-CLM initialized with the bounding box from a multi-task cascaded neural network face detector.

Other tracking methods perform face detection only in the first frame and then apply facial landmark localization using the fitting result from the previous frame as initialization. The estimated head position is expressed in terms of the position of the head with respect to the camera in millimeters (Tx, Ty, and Tz) and rotations in radians around the x, y, and z axes (Rx, Ry, Rz). The 18 recognized facial action units are expressed in terms of presence (0–1) and intensity (on a 6-pointLikert scale).

#### 2.4.2. Classification

We performed intra-subject validation using raw data [3]. In the intra-subjective case, the classification wascarried out on the characteristics of each participant individually. To minimize bias, the 10-cross fold validation technique wasapplied to each participant dataset. Using the raw representation, the data wasrepresented by a vector of numbers with real values.

The z-normalization wasapplied on vectors. According to the IAPS picture (positive, negative, and neutral) table provided by the expert, each instance wastagged manually.

Attitude assessment wascarried out using the following algorithms:KNN—a non-parametric algorithm employed for regressions and classification. Class membership of each point is calculated by a majority vote of the closest neighbors of each point [3]: a query point is assigned the data class that has the greatest number of representatives within the point’s closest neighbors [3]. In our case, we used K = 3 [3].RF—an ensemble learning technique that works by building multiple decision trees during training [3]. The class mode reverts to the corresponding class [3]. We set up a maximum of 64 trees in the forest and the entropy function to measure the split quality [3].

For these methods, we used the sklearn Python toolbox for machine learning [3,41] in the Google Collaboratory cloud service [3,42]. The effectiveness of each algorithm wasestimated in terms of accuracy (A), precision (P), recall (R), F-Measure (F), and execution time (T) [3]. The same metrics werealso used to compare the performance of the two algorithms in analyzing the dataset [3].

### 2.5. Statistical Analysis

Data analyses were performed using R Ver. 4.1.2. Statistical software package (The R Project for Statistical computing; available at URL http://www.r-project.org/; Accessed 18 November 2021). For dichotomous variables, differences between the groups were tested using the Fisher exact test. This analysis was made using the two-way contingency table analysis available at the Interactive Statistical Calculation Pages (http://statpages.org/; Accessed 18 November 2021). For continuous variables, normal distribution was verified by the Shapiro–Wilk normality test and the 1-sample Kolmogorov–Smirnov test. For normally distributed variables, differences among the groups were tested by the Welch two-sample *t*-test or analysis of variance under general linear model. For non-normally distributed variables, differences among the groups were tested by the Wilcoxon rank sum test with continuity correction or the Kruskal–Wallis rank sum test. Test results in which the *p*-value was smaller than the type 1 error rate of 0.05 were declared significant.

## 3. Results

### 3.1. Participant Characteristics

Of all participants, three (users n. 10, 15, and 25) were excluded because the videos recorded were not properly visible. In total, 27 participants (M = 12, F = 15, mean age = 40.48 ± 10.82 years) were included in the study according to the inclusion and exclusion criteria as shown in Table 3. Of these 27, 18 participants used the robot in the static modality while 9 participants used the robot in the active modality.

The two groups did not differ for gender (*p* = 0.411) and educational level (*p* = 0.194). The participants who used the robot in static modality were younger (*p* = 0.049) than the other group.

### 3.2. Emotion Analysis

The longest interaction session lasted 29 min while the shortest one lasted 17 min. Figure 4 describes the total frames for each participant. In intra-subject validation, the number of samples in the training and testing dataset depends on the number of frames recorded for each user.

As reported in Figure 5 and Figure 6, dealing with the raw representation, the KNN obtainedan average precision of 0.85 ± 0.06 and anaverage speed of the algorithm of 6.62 ± 1.62 s.

The performance of RF algorithm wasbetter in terms of accuracy (mean ± sd = 0.98 ± 0.01) and execution time (mean ± sd = 5.73 ± 0.86 s) with respect to KNN algorithm. By KNN algorithm, the neutral attitudes showed the worst performance in terms of precision (mean ± sd = 0.82 ± 0.07) and F-measure (mean ± sd = 0.84 ± 0.06) followed bypositive and negative attitudes. According to RF algorithm, all neutral, positive, and negative attitudes had an equal and high precision (mean = 0.98) and F-measure (mean = 0.98). The participant groups that used the robot in static and active modalities did not differ in all considered variables according to KNN and RF algorithms.

### 3.3. Usability and Acceptability Results

As shown in Table 4, the three groups of participants did not differ in SUS (*p* = 0.157) and AMQ domains (ANX, *p* = 0.716; ATT, *p* = 0.726; FC, *p* = 0.226; ITU, *p* = 0.525; PAD, *p* = 0.701; PENJ, *p* = 0.624; PEOU, *p* = 0.525; PS, *p* = 0.527; PU, *p* = 0.519; SP, *p* = 0.197; SI, *p* = 0.194). Most participants confirmed a high level of usability of the robot solution (SUS > 70 in mean) and acceptability as shown in the AMQ domains: low level of anxiety (ANX = 7.59 in mean), good attitude (ATT = 11.59 in mean), few facilitating conditions (FC = 6.18 in mean), high level of intention to use (ITU = 8.44 in mean), high level of perceived adaptability (PAD = 10.74 in mean), highest level of perceived enjoyment (PENJ = 20.18 in mean), highest level of perceived ease of use (PEOU = 16.96 in mean), high level of perceived sociability (PS = 13.74 in mean), high level of perceived usefulness (PU = 9.85 in mean), and highest level of social presence (SP = 14.44 in mean). Only the social influence level was low (SI = 5.48 in mean) because the experiment did not involve connecting with other people.

## 4. Discussion

In this study, the main objective was to explore if automatic tools couldimprove emotion detection and the robot couldbe a disturbing factor in the elicitation of emotions. The outcomes showedthat extracted visual features madethe attitude assessment stronger. In this study, no sufficient audio data belonged to each attitude state to investigate this modality on its own. As for the intra-subject analysis, RF achieved high performance in terms of accuracy for the raw representation. As shown in [43], the results obtained in the intra-subject validation showedthat the assessment can be customized for each user, suggesting the possibility of integrating this representation into a larger architecture.

Conversely, a recent studydemonstrated that KNN algorithm had a superior performance as well as the support vector machine (SVM) and multilayer perceptron (MLP) algorithms [27]. In another study, among ten machine learning algorithms applied to predict four types of emotions, RF showed the highest area under the curve of 0.70, thenother algorithms including KNN (0.61) [44]. There are discrepancies between the results of previous studies and the present study. Previously, studies reported SVM [45,46,47,48,49] and KNN [50,51] superiority. Possible reasons for these divergences could be the following: (1) SVM has parameters that include C and gamma that can cause overfitting problems (i.e., high accuracy in the training dataset and low accuracy in the test dataset); (2) it is unclear which previous studies used proper cross-validations for their algorithms because they did not describe how to conduct cross-validations; (3) SVM and KNN belong to the classifier type of machine learning algorithms whereas RF belongs to the ensemble type of machine learning algorithms; (4) RF analyzes high-dimensional data and solves a variety of problems to achieve high accuracy [52]: this contrasts with simple classifiers such as SVM and KNN which are suitable for small sample sizes.

The information coming out of encoder includes the most pertinent features identified by the robot’s perception system, decreasing setting assessment times. The flow of information proposed in this studycan be integrated into a cognitive architecture to model robot behavior based on user behavior [3]. Attitude information can be used not only to define what to do but also how the robot should perform the task [3]. This study wascarried out with a relatively large data sample. As a preliminary work, we used common machine learning algorithms to assess the user’s attitude state. Once more data areavailable, more advanced neural architectures can be introduced. We strongly believe that by introducing neural architecture, the robot can automatically assess online what has been done offline.

Furthermore, the fundamental result is that all the participating groups that used the robot in static and active modalities did not differ in any ofthe considered variables according to the KNN and RF algorithms. This aspect canmean that a static or active robot is not a disturbing factor in the arousal of emotions.

Furthermore, most of the participants confirmed a high level of usability and acceptability of the robotic solution.

The outcome of all work can be integrated into a robotic platform to automatically assess the quality of interaction and to modify its behavior accordingly.

Affective computing or social robotics increased not only the necessity of studies about how to improve the emotional interaction between humans and machines but also how to design cognitive architectures which included biomimetic elements related with emotions. The range of emotional aspects with fundamental interest for robot engineers and experts is comprehensive: human–robot interaction, robot task-planning, energy management, social robotics, body design, care robotics, and service robotics, among a long list. Practically, there is no field related to robotic and AI thatis not directly or indirectly related to the implementation of emotional values.

Methods of evaluation for human studies in HRI are listed as (1) self-assessments, (2) interviews, (3) behavioral measures, (4) psychophysiology measures, and (5) task performance metrics, with self-assessment and behavioral measures being the most common.

As the area of social robotics and HRI grows, public demonstrations have the potential to provide insights about the robot and system effectiveness in public settings and reactions of the people. Live public demonstrations enable us to better understand humans and inform the science and engineering fields to design and build better robots with more purposeful interaction capabilities.

One of the challenges is that, although modelling the dynamics of expressions and emotions has been extensively studied in the literature, how to model personality in a time-continuous manner has been an open problem.

Robust facial expression recognition is technically challenging, especially if the age range of the intended participants is large; the facial expression dataset of this thesis contained only adult participants. The technical challenges are compounded by the fact that expression recognition needs to be carried out with real-time processing speed on a standard computer. It needs to conductmultiple live demonstrations in elderly patients without cognitive impairment and with dementia. This isrequired to build a robust facial expression recognition system.

## 5. Conclusions

The study aimed to identify if traditional machine learning algorithms (KNN and RF) could be used to assess three types of emotion valences (positive, negative, andneutral) for every user, to relate emotion recognizing in two robotic modalities (static or motion robot), and to evaluate the acceptability and usability of assistive robot from an end-user point of view. According to the analysis of emotions performed on the recorded videos, RF algorithm performance wasbetter in terms of accuracy and execution time than KNN algorithm. Most of the participants confirmed a high level of usability and acceptability of the robotic solution.

In conclusion, the robot wasnot a disturbing factor in the arousal of emotions.

## Figures and Tables

**Figure 1 sensors-22-02861-f001:**
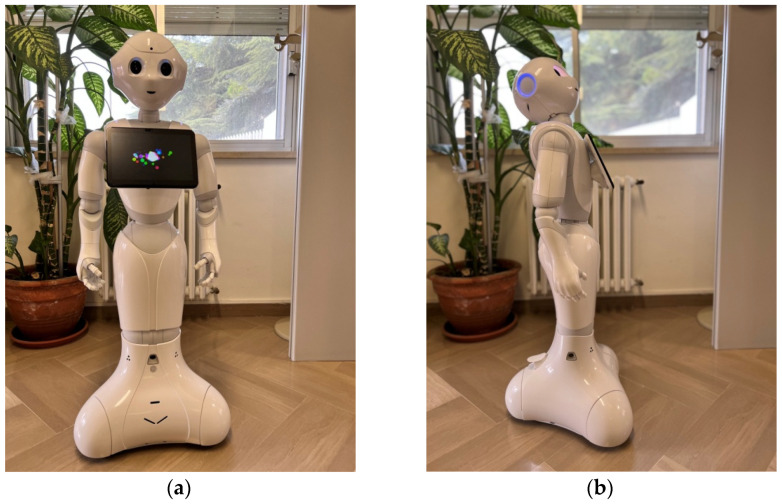
Overview of Pepper robot in front view (**a**) and side view (**b**).

**Figure 2 sensors-22-02861-f002:**
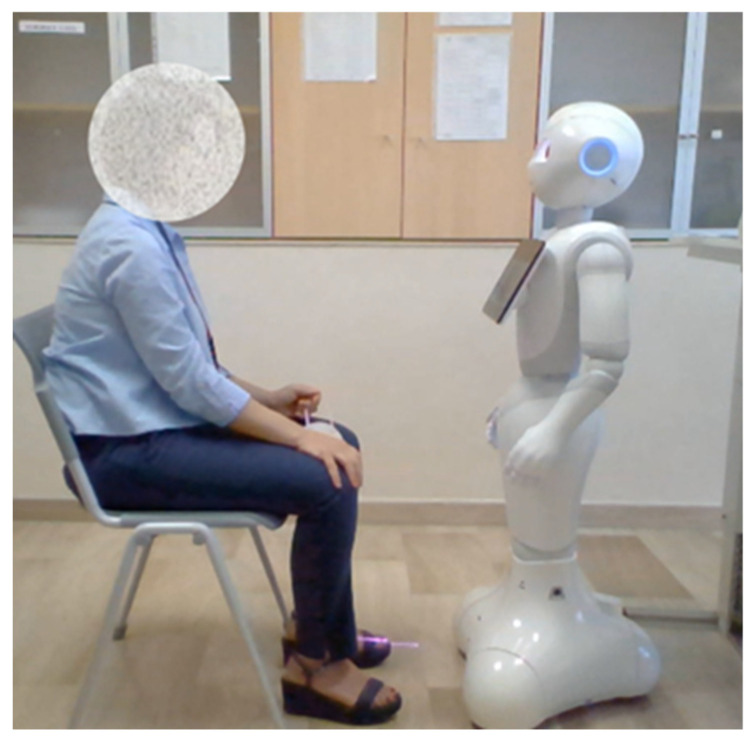
Overview of experimental context.

**Figure 3 sensors-22-02861-f003:**
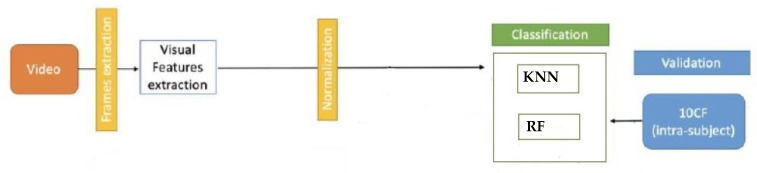
Data analysis.

**Figure 4 sensors-22-02861-f004:**
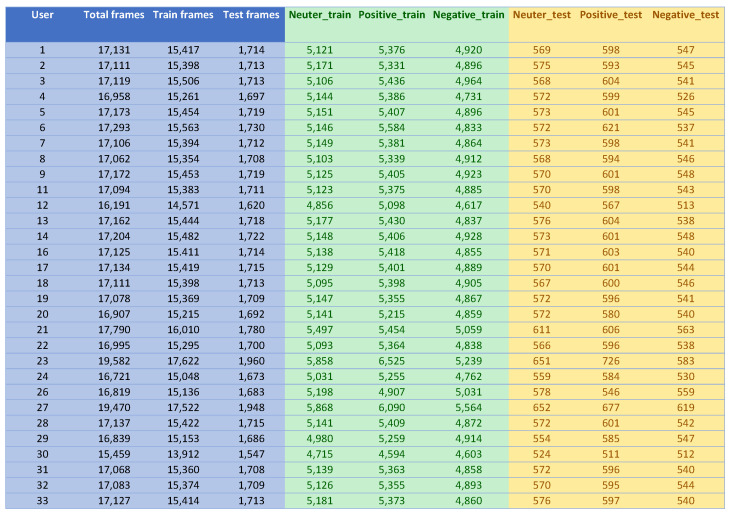
Total frames analyzed for each participant.

**Figure 5 sensors-22-02861-f005:**
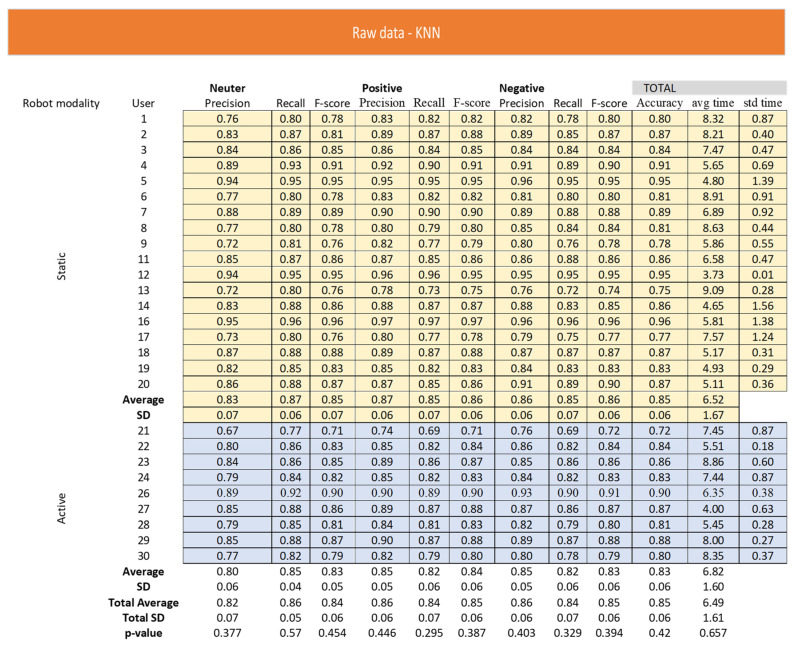
Raw representation according to KNN algorithm.

**Figure 6 sensors-22-02861-f006:**
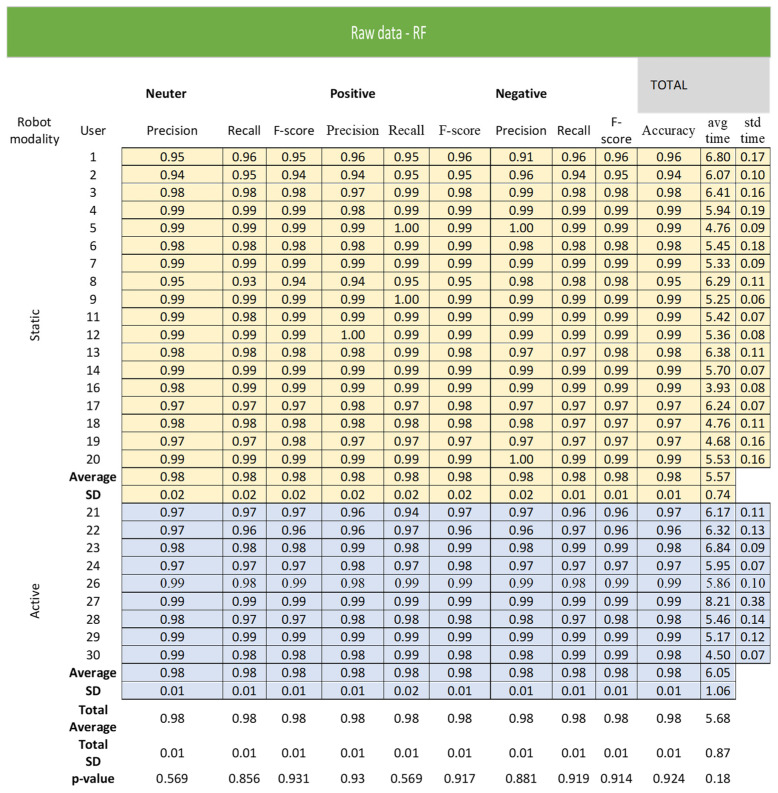
Raw representation according to RF algorithm.

**Table 1 sensors-22-02861-t001:** Robot actions performed according to positive or negative images shown.

Positive	Negative
To smile	To step back slightly showing disgust
To clap hands	To cry
To raise arms and cheer	To bend chest forward showing boredom
To blow a kiss	To turn head left and right quickly showing fear
To wave	To bow head showing sadness
To make an appreciation	To fold arms showing confusion

**Table 2 sensors-22-02861-t002:** Social cues analyzed.

Parameter	Category	Types
Behavioral	**Emotion**	Joy, sadness, fear, anger, disgust, neutral
	**Gaze**	Directed gaze, mutual face gaze, none
	**Facial expressions**	Smile, laugh, raise eyebrows, frown, inexpressive

**Table 3 sensors-22-02861-t003:** Participant characteristics.

	All *n* = 27	Static Robot *n* = 18	Accordant Motion *n* = 9	*p*-Value
Gender				0.411
Men/Women	12/15	7/11	5/4	
Men (%)	44.40	38.90	55.60	
Age (years)				**0.049**
Mean ± SD	40.48 ± 10.82	37.61 ± 8.14	46.22 ± 13.56	
Range	28–66	28–53	31–66	
Educational level				0.194
Degree—*n* (%)	24 (88.90)	17 (94.40)	7 (77.80)	
High school—*n* (%)	3 (11.10)	1 (5.60)	2 (22.20)	

**Table 4 sensors-22-02861-t004:** Usability and acceptability post-robot interaction.

	All *n* = 27	Static Robot *n* = 18	Accordant Motion *n* = 9	*p*-Value
**SUS**				0.157
Mean ± SD	72.87 ± 13.11	75.42 ± 14.98	67.78 ± 6.18	
Range *	45.00–100.00	45.00–100.00	60.00–77.50	
**AMQ**				0.716
ANX				
Mean ± SD	7.59 ± 2.54	7.62 ± 2.60	7.33 ± 2.54	
Range *	4–13	4–13	4–11	
ATT				0.726
Mean ± SD	11.59 ± 1.88	11.50 ± 2.01	11.78 ± 1.71	
Range *	7–15	7–15	9–14	
FC				0.226
Mean ± SD	6.18 ± 1.88	6.50 ± 2.09	5.56 ± 1.24	
Range *	2–10	2–10	4–8	
ITU				0.525
Mean ± SD	8.44 ± 3.13	8.72 ± 3.18	7.89 ± 3.14	
Range *	3–15	3–15	3–12	
PAD				0.701
Mean ± SD	10.74 ± 1.72	10.83 ± 1.85	10.55 ± 1.51	
Range *	7–15	7–15	8–13	
PENJ				0.624
Mean ± SD	20.18 ± 2.97	20.39 ± 3.29	19.78 ± 2.33	
Range *	15–25	15–25	16–24	
PEOU				0.525
Mean ± SD	16.96 ± 2.71	16.72 ± 3.02	17.44 ± 2.01	
Range *	12–21	12–21	14–20	
PS				0.527
Mean ± SD	13.74 ± 2.72	13.50 ± 3.18	14.22 ± 1.48	
Range *	4–18	4–18	12–16	
PU				0.519
Mean ± SD	9.85 ± 2.26	10.05 ± 2.48	9.44 ± 1.81	
Range *	5–15	5–15	7–12	
SI				0.197
Mean ± SD	5.48 ± 2.08	5.11 ± 2.13	6.22 ± 1.85	
Range *	2–9	2–8	4–9	
SP				0.194
Mean ± SD	14.44 ± 2.79	13.94 ± 2.62	15.44 ± 3.00	
Range *	9–19	9–19	9–19	

* Minimum and maximum scores obtained by the participants.

## Data Availability

The data presented in this study are available on request from the corresponding author. The data are not publicly available due to restrictions (their containing information that could compromise the privacy of research participants).

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
