# Peer review of "Emotion Recognizing by a Robotic Solution Initiative (EMOTIVE Project)"

_sensors, 2022, doi:10.3390/s22082861_

Round 1

Reviewer 1 Report

The article “EMotion recognizing by a rOboTIc solution initiatiVE (EMOTIVE) project” presents a system to recognize emotions in adults when they are seeing images to elicitate emotions rated previously and available in a data set called IAPS. These images are supposedly to cause neutral, positive and negative emotions. Proposed system compares use k-nearest neighbor (KNN) and Random Forest (RF) to classify emotions using face images captured when user is seeing the pictures from IAPS and a robot is being used in parallel in static or active mode.

The work presented is interesting, the article is ease to read, but it is missing some important information to allow a better understanding.

In the following will be presented points to be corrected or information to be complemented.

At the abstract and in the text is not clear the rule of the robot in the experiments. At Figure 1, there is a photography of the robot with a tablet in the chest. Are the images presented in this tablet to the user?

Results and conclusions at the abstract reuse a phrase that does not clarify what is the best method in the comparison. Reading the text is possible to know that RF has better results that KNN but it is not clear in the abstract.

At the Introduction, paragraph between lines 104 to 109 uses the acronym SOM without presentation, and uses ‘he’ for the robot. All text uses ‘it’ for the robot.

At lines 118 and 119 the sentence “Interestingly, emotion recognition in robotics is important for several reasons, not only for the pure interaction between human robots, but also for bioengineering operations.” What is a human robot?

At line 162 is presented a list of six items, but there is a problem with numbering them.

At line 184 is presented a list of three items, but there is also a problem with the numbers. Basically, the first one has not number.

At line 194, the last criteria of exclusion should start with Capital letter.

Experimental protocol is not complete. It is missing the information about kind of movements robot executes in its active mode. It is saying that “About the experimental phase with Pepper in active modality, a concordant mimic was programmed according to the image types (positive, negative and neutral).” Are the movements dependent of the image or only the emotion expressed?

Experimental protocol is missing information about positioning of robot and user.

Where is the tablet in which the user sees pictures of IAPS?

Does the user evaluate pictures using SAM in the tablet or in a piece of paper?

Does the user have time extra to realize evaluation between pictures or does the evaluation takes place at the same time interval of 7 seconds in which the picture is shown?

Is the user required to stand or sit in the session? How much time experiments last? Are all 60 pictures shown to each user? Are they shown in a specific order? Do all users see the pictures at the same order?

It is necessary explain how “the Almere Model Questionnaire (AMQ) [37] and System Usability Scale (SUS) [38]” works without the need to read another paper to understand. For example, it is important to the reader to know that SUS has a mean of 68, so values above that indicate that the system is useful and easy to use.

At the Results there is information about the experimental protocol: “In total, 27 participants (M = 12, F = 15, mean age = 40.48± 10.82 years) were included in the study, according to the inclusion/exclusion criteria as shown in Table 2, and randomly assigned to two groups according to static and active modalities of the robot: 18 participants had used the robot in static modality, and 9 participants had used the robot in active modality.”

The sentence at lines 313 and 314 is confusing: “The participants who had used the robot in static modality were younger (p = 0.049) than other two groups.” Are there three groups? Because one group is younger than other two groups!!

At session Results, it was changed the way to refer to the emotions represented in images from “neutral” to “neuter”. Use neutral at all the text.

At lines 331 to 333 the sentence does not pass the correct conclusion that RF is better than KNN: “According to RF algorithm, all neutral, positive and negative attitudes had an equal and high precision (mean = 0.98) and F-measure (mean = 0.98), with respect to KNN algorithm.”

Results for AMQ and SUS at session 3.3 Usability and Acceptability Results are impossible to understand reading only this paper. It has a great number of acronyms explained only as legend in a table. What is a good value for each item?

At the legend of Table 3 there are words inappropriately connected.

At line 368 “relatively data large sample” should be changed for “relatively large data sample.

Author Response

  • The article “EMotion recognizing by a rOboTIc solution initiatiVE (EMOTIVE) project” presents a system to recognize emotions in adults when they are seeing images to elicitate emotions rated previously and available in a data set called IAPS. These images are supposedly to cause neutral, positive and negative emotions. Proposed system compares use k-nearest neighbor (KNN) and Random Forest (RF) to classify emotions using face images captured when user is seeing the pictures from IAPS and a robot is being used in parallel in static or active mode. The work presented is interesting, the article is ease to read, but it is missing some important information to allow a better understanding. In the following will be presented points to be corrected or information to be complemented.
  • We thank the reviewer for his/her positive and constructive observations. Below, please find item-by-item responses to your comments, which are included verbatim.

  • At the abstract and in the text is not clear the rule of the robot in the experiments. At Figure 1, there is a photography of the robot with a tablet in the chest. Are the images presented in this tablet to the user?
  • Yes, the pictures presented by the tablet mounted on the robot chest. We added this information in the Abstract section and “2.4. Experimentation protocol” (Material and Methods section).

  • Results and conclusions at the abstract reuse a phrase that does not clarify what is the best method in the comparison. Reading the text is possible to know that RF has better results that KNN but it is not clear in the abstract.
  • According to reviewer observation, we added the following sentence in the Abstract section: “…RF algorithm performance is better in terms of accuracy (mean±sd=0.98±0.01), and execution time (mean±sd=5.73±0.86 sec) than KNN algorithm”.

  • At the Introduction, paragraph between lines 104 to 109 uses the acronym SOM without presentation, and uses ‘he’ for the robot. All text uses ‘it’ for the robot.
  • We added “Self-Organizing Map” to explain the acronym SOM, and correct “it” instead of “he”.

  • At lines 118 and 119 the sentence “Interestingly, emotion recognition in robotics is important for several reasons, not only for the pure interaction between human robots, but also for bioengineering operations.” What is a human robot?
  • We correct the typo as “…between human and robot”.

  • At line 162 is presented a list of six items, but there is a problem with numbering them.
  • We correct the list numbering.

  • At line 184 is presented a list of three items, but there is also a problem with the numbers. Basically, the first one has not number.
  • We correct the list numbering.

  • At line 194, the last criteria of exclusion should start with Capital letter.
  • We correct the typo.

  • Experimental protocol is not complete. It is missing the information about kind of movements robot executes in its active mode. It is saying that “About the experimental phase with Pepper in active modality, a concordant mimic was programmed according to the image types (positive, negative and neutral).” Are the movements dependent of the image or only the emotion expressed?
  • According to reviewer observation, we added the following sentence in order to clarify the programmed movements of the robot: “…the robot performed an action according to the valence of each image shown (i.e., it smiled if a picture of a smiling child was shown)”.

  • Experimental protocol is missing information about positioning of robot and user. Where is the tablet in which the user sees pictures of IAPS? Does the user evaluate pictures using SAM in the tablet or in a piece of paper? Does the user have time extra to realize evaluation between pictures or does the evaluation takes place at the same time interval of 7 seconds in which the picture is shown? Is the user required to stand or sit in the session? How much time experiments last? Are all 60 pictures shown to each user? Are they shown in a specific order? Do all users see the pictures at the same order?
  • The tablet was on robot chest and in front of the user. The user evaluated pictures using SAM directly in the tablet. Only each picture was shown for 7 seconds; the user had plenty of time to realize evaluation. The user was sitting during the session. The experiment lasted about 20-25 minutes for participant. All 60 pictures showed to each user. The pictures did not follow a specific layout, but they was shown in the same order to all participants. In order to clarify the experimental protocol, we added a figure (Overview of experimental context) and the aforesaid information in the text.

  • It is necessary explain how “the Almere Model Questionnaire (AMQ) [37] and System Usability Scale (SUS) [38]” works without the need to read another paper to understand. For example, it is important to the reader to know that SUS has a mean of 68, so values above that indicate that the system is useful and easy to use.
  • According to reviewer observation, we added an explanation about AMQ and SUS works.

  • At the Results there is information about the experimental protocol: “In total, 27 participants (M = 12, F = 15, mean age = 40.48 ± 10.82 years) were included in the study, according to the inclusion/exclusion criteria as shown in Table 2, and randomly assigned to two groups according to static and active modalities of the robot: 18 participants had used the robot in static modality, and 9 participants had used the robot in active modality.”
  • We added the following information to “2.4. Experimentation protocol” section: “The participants were randomly assigned to one of two groups according to static (Static robot group) and active (Accordant Motion group) modalities of the robot”. We deleted “… and randomly assigned to two groups according to static and active modalities of the robot:” in Results section.

  • The sentence at lines 313 and 314 is confusing: “The participants who had used the robot in static modality were younger (p = 0.049) than other two groups.” Are there three groups? Because one group is younger than other two groups!!
  • We correct the typo.

  • At session Results, it was changed the way to refer to the emotions represented in images from “neutral” to “neuter”. Use neutral at all the text.
  • We substituted “neutral” to “neuter”, according to reviewer suggestion.

  • At lines 331 to 333 the sentence does not pass the correct conclusion that RF is better than KNN: “According to RF algorithm, all neutral, positive and negative attitudes had an equal and high precision (mean = 0.98) and F-measure (mean = 0.98), with respect to KNN algorithm.”
  • The aforesaid sentence refers to outcomes achieved by RF algorithm. Therefore the comparison with KNN is not required. For this reason, we deleted “…with respect to KNN algorithm”.

  • Results for AMQ and SUS at session 3.3 Usability and Acceptability Results are impossible to understand reading only this paper. It has a great number of acronyms explained only as legend in a table. What is a good value for each item?
  • We clarified AMQ and SUS features in the Material and Methods section.

  • At the legend of Table 3 there are words inappropriately connected.
  • We correct the typo.

  • At line 368 “relatively data large sample” should be changed for “relatively large data sample.
  • We correct the sentence according to reviewer suggestion.

Reviewer 2 Report

The authors provided a manuscript on a very actual topic. The manuscript is quite straightforward and well organized. Nevertheless, I noticed some issues to be resolved.

  1. In the title, the authors refer to the project by using some capital letters. In my opinion, it results in more efforts to realize the idea. I recommend avoiding the use of additional capital letters in the title. Also, the word „project“at the end of the title looks strange. I recommend clarifying the title to avoid possible misunderstandings.
  2. The abstract is very detailed from my point of view. It looks a little bit too extensive.
  3. Line 105, abbreviation SOM not explained; line 167 what means inertial unit? Maybe it is an inertial measurement unit (IMU)?
  4. Is it worth doing such small subchapters as 2,2, and 2,3? Maybe the information can be added to the body text?
  5. I recommend including a chapter for conclusions and providing some concentrated statements supported by obtained quantitative results.

Author Response

  • The authors provided a manuscript on a very actual topic. The manuscript is quite straightforward and well organized. Nevertheless, I noticed some issues to be resolved.
  • We thank the reviewer for his/her positive and constructive observations. Below, please find item-by-item responses to your comments, which are included verbatim.

  • In the title, the authors refer to the project by using some capital letters. In my opinion, it results in more efforts to realize the idea. I recommend avoiding the use of additional capital letters in the title.
  • According to reviewer observation, we changed the additional capital letters to lowercase letters.

  • Also, the word „project“ at the end of the title looks strange. I recommend clarifying the title to avoid possible misunderstandings.
  • We included the word “project” in parenthesis with acronym “EMOTIVE”, as shown below: “Emotion recognizing by a robotic solution initiative (EMOTIVE project)”.

  • The abstract is very detailed from my point of view. It looks a little bit too extensive.
  • According to reviewer observation, we improved the Abstract section in terms of unnecessary detail reduction.

  • Line 105, abbreviation SOM not explained; line 167 what means inertial unit? Maybe it is an inertial measurement unit (IMU)?
  • We added “Self-Organizing Map” to explain the acronym SOM, and correct the typo about IMU.

  • Is it worth doing such small subchapters as 2,2, and 2,3? Maybe the information can be added to the body text?
  • We merged the two aforesaid subchapters into a single subchapter entitled “Recruitment”.

  • I recommend including a chapter for conclusions and providing some concentrated statements supported by obtained quantitative results.
  • We added the Conclusion section as well as recommended by the reviewer.

Reviewer 3 Report

This paper evaluates emotion recognition using a humanoid robot as a disturbing factor. A set of pictures with positive/negative/neutral valence are displayed on a screen located on the robot's body while this is static or in motion. A group of 30 voluntary subjects reported the emotional experience while being recorded. The videos are further analyzed with classic machine learning algorithms such as KNN and Random Forest to recognize the users' emotions. The experiment is completed by an acceptability survey by the participants.

Overall, the paper addresses a topic on social robots, which could find application in methods to improve the quality of human-robot interactions. I am recommending a major revision with the following remarks:

1. The paper needs a deep language revision. The paper is not that fluent to read due the number of grammatical flaws in it.

2. Line 216: please describe and illustrate the concordance mimic performed by the robot according to the three image types.

3. It is not clear if videos are recorded with the robot's inbuilt cameras or with an external one.

4. The CE-CLM architecture should be described. As it is, it seems a black box inside the OpenFace toolkit.

5. What is the point of Figure 3? A text stating than n frames are needed for the training stage and m for the tests should be enough.

6. A graphic representation of figures 4 and 5 is more suitable to present the data.

Author Response

  • This paper evaluates emotion recognition using a humanoid robot as a disturbing factor. A set of pictures with positive/negative/neutral valence are displayed on a screen located on the robot's body while this is static or in motion. A group of 30 voluntary subjects reported the emotional experience while being recorded. The videos are further analyzed with classic machine learning algorithms such as KNN and Random Forest to recognize the users' emotions. The experiment is completed by an acceptability survey by the participants. Overall, the paper addresses a topic on social robots, which could find application in methods to improve the quality of human-robot interactions. I am recommending a major revision with the following remarks.
  • We thank the reviewer for his/her positive and constructive observations. Below, please find item-by-item responses to your comments, which are included verbatim.

  • The paper needs a deep language revision. The paper is not that fluent to read due the number of grammatical flaws in it.
  • According to reviewer suggestion, we obtained a review of the manuscript from a native English speaker.

  • Line 216: please describe and illustrate the concordance mimic performed by the robot according to the three image types.
  • According to reviewer observation, we improved the experimentation protocol section and added the following sentence in order to clarify the programmed movements of the robot: “…the robot performed an action according to the valence of each image shown (i.e., it smiled if a picture of a smiling child was shown)”.

  • It is not clear if videos are recorded with the robot's inbuilt cameras or with an external one.
  • The videos are recorded with an external camera. In order to clarify this aspect of the experimentation sessions that the reviewer has rightly pointed out, we added the aforesaid information in the text.

  • The CE-CLM architecture should be described. As it is, it seems a black box inside the OpenFace toolkit.
  • CE-CLM architecture description has been added as well as the reviewer recommended.

  • What is the point of Figure 3? A text stating than n frames are needed for the training stage and m for the tests should be enough.
  • Actually Figure 4 describes the total frames for each participant. Even if the reviewer has rightly highlighted the complexity of the figure, we think it useful to leave the figure to better visualize the number of frames recorded for each user.

  • A graphic representation of figures 4 and 5 is more suitable to present the data.
  • Currently Figure 5 and Figure 6 show the raw representation according to KNN and RF algorithms. Also in this case, we prefer to leave the tables as they are to better represent the collected data, in the hope that the reviewer understands our reasons.

Reviewer 4 Report

The paper presents an interesting subject - the following aspects must be clarified:

  • it is not clear the novelty of the paper; what is the role of the robot? Is it necessary? How can the robot can be included in the human computer interaction scenario?
  • a state of the art section must be added - containing other existing works together with obtained results; explain the selected methods for performing emotion recognition (based on other solutions)
  • references must be updated - update with references from the last five years
  • what was the inference time: acquisition time and processing time (for obtaining emotion)
  • examples of the dataset must be added; what does it mean a positive / a negative emotion? why the proposed method was not tested on other existing datasets? why only 3 classes of emotions were used (and not more classes like in other existing methods)?
  • what are the performances in case of different environment conditions, eg. low or high illumination conditions (for the user)?
  • comparison with other methods must be added

Author Response

  • The paper presents an interesting subject - the following aspects must be clarified: it is not clear the novelty of the paper; what is the role of the robot? Is it necessary? How can the robot can be included in the human computer interaction scenario?
  • We thank the reviewer for his/her positive and constructive observations. Below, please find item-by-item responses to your comments, which are included verbatim. Novelty the paper resides in the use of a robot for an emotional detection experimentation. A fundamental objective was     to compare emotion recognizing in two types of robotic modality: static robot (which does not perform any movement) and motion robot (who performs movements in concordant with the emotions elicited).

  • A state of the art section must be added - containing other existing works together with obtained results; explain the selected methods for performing emotion recognition (based on other solutions).
  • According to reviewer suggestion, we improved the Introduction section about recent studies focused on emotion recognition methods.

  • References must be updated - update with references from the last five years.
  • Except for references 1, 2 and 5 which are the theoretical basis of the study of emotions, we have updated the bibliography from 2014 to today.

  • What was the inference time: acquisition time and processing time (for obtaining emotion).
  • In order to improve the experimental protocol section, we added a figure (Overview of experimental context) and further information in the text.

  • Examples of the dataset must be added; what does it mean a positive / a negative emotion? why the proposed method was not tested on other existing datasets? why only 3 classes of emotions were used (and not more classes like in other existing methods)?
  • According to reviewer observations, we clarify the emotion type meanings in the “2.3. Experimentation Protocol” section. We used the three valences of emotions as theoretically described by Ekman.

  • What are the performances in case of different environment conditions, eg. low or high illumination conditions (for the user)?
  • This bias has avoided because the experiment was performed in the same setting with the same lighting for all participants.

  • Comparison with other methods must be added.
  • Comparison with other methods and studies has been added in Discussion section, as well as suggested by the reviewer.

Round 2

Reviewer 1 Report

The article “Emotion recognizing by a robotic solution initiative (EMOTIVE project)” in this version has a presentation of protocol in details and explains better measures used to compare solutions. Sentences are also written in a more connected way, easier to read.

In the following there are some points that should be corrected.

At the abstract (line 38) the conclusion has still an affirmation that does not says what it is proposed to: “Conclusions: RF algorithm, all neutral, positive and negative attitudes had equal and high precision and F measurement, compared to the 39 KNN algorithm.” It is recommended to rewrite it.

At section 2.3 presenting the protocol, it was great having information about what kind of movement robot executes in active mode in reaction to the emotion detected. However, it should be clearer if the robot always changes its emotion expression at the face or it can realize another movement, like clapping hands to show happiness. It should be great having a list of possible movements realized for the robot. Considering that it would be more interesting if reaction is not the same any time in human, and that the user could feel boring about repetitive robot reaction.

Table 1 at line 281 has one line with different space configuration. All lines should have the same dimensions but they do not.

Table 1 is repeated at line 411 with more configuration problems.

Table 2 is presented at line 362 and repeated at line 412.

Table 3 presents a range for each item. Is it the theoretical range in the scale? It should be informed in the Legend to ease understanding.

At line 435 when comparing results with other works, it is used a new acronym AUC that is not presented before.

At lines 440, 441 e 442 “2) previous studies used proper cross-validations for their algorithms is unclear, because some of the studies did not describe how to conduct cross-validations;” It should be rewritten because it is confusing. May be: “2) previous studies that used proper cross-validations for their algorithms are unclear, because they did not describe how to conduct cross-validations;”

Typing mistakes:

Insert a space after “:” at line 41.

Put “Otherwise” in capital letter at line 123.

Change “a” for “an alternate case” at line 127.

At line 136 change “used” for “was used”

At line 144 change “classifiers was learned” for “classifiers were learned”.

At line 185 exclude “Recuitment” in the phrase “RecruitmentParticipants were recruited on”

Author Response

  1. The article “Emotion recognizing by a robotic solution initiative (EMOTIVE project)” in this version has a presentation of protocol in details and explains better measures used to compare solutions. Sentences are also written in a more connected way, easier to read. In the following there are some points that should be corrected.                                                      We thank the reviewer for his/her positive and constructive observations. Below, please find item-by-item responses to his/her comments, which are included verbatim.
  2. At the abstract (line 38) the conclusion has still an affirmation that does not says what it is proposed to: “Conclusions: RF algorithm, all neutral, positive and negative attitudes had equal and high precision and F measurement, compared to the 39 KNN algorithm.” It is recommended to rewrite it.                                                                                          According to the reviewer suggestion, we rewrite the sentence as shown below: “RF algorithm performance is better in terms of accuracy and execution time than KNN algorithm”.

  1. At section 2.3 presenting the protocol, it was great having information about what kind of movement robot executes in active mode in reaction to the emotion detected. However, it should be clearer if the robot always changes its emotion expression at the face or it can realize another movement, like clapping hands to show happiness. It should be great having a list of possible movements realized for the robot. Considering that it would be more interesting if reaction is not the same any time in human, and that the user could feel boring about repetitive robot reaction. We added a new table (Table 1) in which the robot actions were shown. The actions of the robot varied in order to avoid the repetition of the same movement in close proximity and to bore the user.

  1. Table 1 at line 281 has one line with different space configuration. All lines should have the same dimensions but they do not.                        Currently, Table 2. We correct the space configuration.

  1. Table 1 is repeated at line 411 with more configuration problems.           The table has been correct.

  1. Table 2 is presented at line 362 and repeated at line 412.                Currently Table 3. The typo has been correct.

  1. Table 3 presents a range for each item. Is it the theoretical range in the scale? It should be informed in the Legend to ease understanding.  Currently Table 4. The reported range shows the minimum and maximum scores obtained by the participants for each item. We added this information below the table as suggested by the reviewer.

  1. At line 435 when comparing results with other works, it is used a new acronym AUC that is not presented before.                                               We correct the acronym with “area under the curve”.

  1. At lines 440, 441 e 442 “2) previous studies used proper cross-validations for their algorithms is unclear, because some of the studies did not describe how to conduct cross-validations;” It should be rewritten because it is confusing. May be: “2) previous studies that used proper cross-validations for their algorithms are unclear, because they did not describe how to conduct cross-validations;”.                                                            We replaced the aforesaid sentence with that proposed by the reviewer.

  1. Typing mistakes:

Insert a space after “:” at line 41.

Put “Otherwise” in capital letter at line 123.

Change “a” for “an alternate case” at line 127.

At line 136 change “used” for “was used”

At line 144 change “classifiers was learned” for “classifiers were learned”

At line 185 exclude “Recuitment” in the phrase “RecruitmentParticipants were recruited on”.

According to reviewer observations, we made the required corrections.

Reviewer 3 Report

The paper has undoubtedly improved from its previous version. It includes now a more detailed description of all the concepts involved.  In particular, I appreciate the clarifications on the CE-CLM architecture and the robot’s mimics together with the comparison with other machine learning algorithms in the Discussion Section.

I have no further major comments or remarks. Therefore, I recommend now this paper’s acceptance.

Author Response

  1. The paper has undoubtedly improved from its previous version. It includes now a more detailed description of all the concepts involved. In particular, I appreciate the clarifications on the CE-CLM architecture and the robot’s mimics together with the comparison with other machine learning algorithms in the Discussion Section. I have no further major comments or remarks. Therefore, I recommend now this paper’s acceptance.
  1. We thank the reviewer for his/her positive comment.

Reviewer 4 Report

Some of my comments were addressed, but there are some that still needs improvements:

  • state of the art must contain also results not only list of methods
  • comparison with other existing methods must be performed also with other existing ones (for example: methods based on convolutional neural networks)

Author Response

  1. Some of my comments were addressed, but there are some that still needs improvements: state of the art must contain also results not only list of methods. Comparison with other existing methods must be performed also with other existing ones (for example: methods based on convolutional neural networks)
  1. We thank the reviewer for his/her positive and constructive comment. According to reviewer observation, we improved state of the art with data and comparisons.